# Membrane metalloendopeptidase (MME) is positively correlated with systemic lupus erythematosus and may inhibit the occurrence of breast cancer

**Jiatong Ding[1,2], Chenxi Li[2], Kexin Shu[2,3], Wanying Chen[2,4], Chenxi Cai[2,4], Xin Zhang[1], Wenxiong Zhang[1]** *

**1** Department of Thoracic Surgery, The Second Affiliated Hospital of Nanchang University, Nanchang, China, **2** Jiangxi Medical College, Nanchang University, Nanchang, China, **3** Department of Rheumatology and Immunology, The Second Affiliated Hospital of Nanchang University, Nanchang, China, **4** Department of Brest Surgery, The second affiliated hospital of Nanchang University, Nanchang, China

* zwx123dr@126.com

## Abstract

### Background

Patients with systemic lupus erythematosus (SLE) have a lower risk of breast cancer (BRCA) than the general population. In this study, we explored the underlying molecular mechanism that is dysregulated in both diseases.

### Methods

Weighted gene coexpression network analysis (WGCNA) was executed with the SLE and BRCA datasets from the Gene Expression Omnibus (GEO) website and identified the potential role of membrane metalloendopeptidase (MME) in both diseases. Then, Gene Ontology (GO) and Kyoto Encyclopedia of Genes and Genomes (KEGG) enrichment analyses of related proteins and miRNAs were performed to investigate the potential molecular pathways.

### Results

WGCNA revealed that MME was positively related to SLE but negatively related to BRCA. In BRCA, MME expression was significantly decreased in tumor tissues, especially in luminal B and infiltrating ductal carcinoma subtypes. Receiver operating characteristic (ROC) analysis identified MME as a valuable diagnostic biomarker of BRCA, with an area under the curve (AUC) value equal to 0.984 (95% confidence interval = 0.976–0.992). KEGG enrichment analysis suggested that MME-related proteins and targeted miRNAs may reduce the incidence of BRCA in SLE patients via the PI3K/AKT/FOXO signaling pathway. Low MME expression was associated with favorable relapse-free survival (RFS) but no other clinical outcomes and may contribute to resistance to chemotherapy in BRCA, with an AUC equal to 0.527 (P value < 0.05).

**Data Availability Statement:** All relevant data are within the manuscript and its Supporting information files.

**Funding:** This study was supported by National Natural Science Foundation of China (NSFC, Grant number: 81560345) and Natural Science Foundation of Jiangxi Province (Grant number: 20212BAB206050). Role of the Funding: The funding had no role in the design and conduct of the study; collection, management, analysis, and interpretation of the data; preparation, review, or approval of the manuscript; and decision to submit the manuscript for publication.

**Competing interests:** The authors have declared that no competing interests exist.

**Abbreviations:** aDC, Activated Dendritic cells; AKT, Protein kinase-b; AUC, Area under curve; BRCA, Breast cancer; CI, Confidence interval; circRNA, Circular RNA; CPTAC, Clinical proteomic tumor analysis consortium; DC, Dendritic cells; ECM, Extracellular matrix; ER, Estrogen receptor; FAK, Focal adhesion kinase; FDR, false discovery rate; follicular Tfh, T follicular helper cells; FOXO, Forkhead box O; GDSC, Genomics of Drug Sensitivity in Cancer; GEO, Gene expression omnibus; GO, Gene ontology; GSCA, Gene set cancer analysis; GSVA, Gene set variation analysis; IC50, 50%inhibiting concentration; iDC, Immature Dendritic cells; KEGG, Kyoto Encyclopedia of Genes and Genomes; Lum B, Luminal-B; M stage, Metastasis stage; miRNA, microRNA; MME, Membrane metalloendopeptidase; mTOR, Mechanistic target of rapamycin; N stage, Node stage; OS, Overall survival; PAM50, prediction analysis of microarray50; pDC, Plasmacytoid Dendritic cells; PI3K, Phosphatidylinositol 3-kinase; PIP2, Phosphatidylinositol 4:5-bisphosphate; PIP3, Phosphatidylinositol 3:4:5-trisphosphate; PPI, Protein-protein interaction; PPS, Post-progression survival; RFS, Relapse-free survival; RhoA, Ras homolog family member A; ROC, Receiver operating characteristic curve; SLE, Systemic lupus erythematosus; STRING, Functional protein association networks; T stage, Tumor stage; TCGA, The cancer genome atlas; Tcm, T central memory; Tgd, T gamma delta cells; Th, T helper cells; WGCNA, Weighted gene co-expression network analysis.

## Conclusions

In summary, MME expression was significantly decreased in BRCA but positively correlated with SLE, and it might reduce the incidence of BRCA in SLE patients via the PI3K/AKT/FOXO signaling pathway.

## Introduction

Breast cancer (BRCA) is recognized as the most common cancer and the leading cause of cancer-related mortality in women worldwide [1]. However, several cohort studies and meta-analyses have revealed that systemic lupus erythematosus (SLE) patients have a decreased risk for BRCA compared with the general population, suggesting a potential protective role of SLE in BRCA [2, 3]. As a devastating chronic autoimmune disease, SLE is a connective tissue disease that affects the risk of several cancers [4]. SLE patients form multiple circulating autoantibodies, including anti-dsDNA antibodies, lupus anticoagulant, and anticardiolipin, which may exert anticancer effects against BRCA [5]. Moreover, BRCA in SLE patients presents a different histologic type and receptor status than that in patients without SLE, with a population-based case–control study reporting that SLE patients have a decreased risk of estrogen receptor (ER)-negative cancers [6]. Thus, we assume that there might be several molecular pathways inhibiting the occurrence of BRCA in SLE patients.

However, few studies have focused on the mechanism of the potential protective role of SLE in BRCA at the genetic level. Due to the rapid development and wide applications of gene microarray technology, large quantities of gene expression data in various diseases have been obtained, which allows us to explore the potential relationship of BRCA and systemic lupus erythematosus at the genetic level. To examine our hypothesis, we performed weighted gene coexpression network analysis (WGCNA) to identify the gene clusters of genes associated with SLE and BRCA.

Consistent with our study, changes in membrane metalloendopeptidase (MME) expression have been identified in several types of cancers. For example, MME is overexpressed in pancreatic endocrine tumors and colorectal carcinoma but decreased in lung and ovarian cancer, suggesting cell type-specific effects of MME [7, 8]. As a type II transmembrane glycoprotein, MME participates in various significant biological processes and usually serves as a tumor suppressor in tumors, such as prostate carcinogenesis and esophageal squamous cell carcinoma [8, 9]. In prostate carcinogenesis, MME often downregulates and attenuates the effects of gastrin-releasing peptide to control the activities of prostate stem/progenitor cells [9]. In esophageal squamous cell carcinoma, MME inhibits the focal adhesion kinase (FAK)- Ras homolog family member A (RhoA) signaling axis to interrupt tumor cell adhesion and metastasis, with a high MME expression level representing a favorable prognosis [8]. However, the literature contains no reports about the effects of MME on BRCA. Therefore, it is crucial to explore the correlation between MME expression and the clinical characteristics of BRCA.

We performed WGCNA with the published gene expression data of SLE and BRCA from the Gene Expression Omnibus (GEO) website. After identifying the potential role of MME in SLE and BRCA, we downloaded the clinical data of BRCA from The Cancer Genome Atlas (TCGA) and further investigated the effects of MME on BRCA.

## Methods

### Acquisition of expression data from GEO

We entered the keywords "breast cancer" or "system lupus erythematosus" in the GEO website to acquire BRCA and SLE gene expression profiles. To ensure the reliability of our results,

suitable datasets were selected with the following criteria: 1) the chosen gene expression profile should contain control and case groups; 2) to confirm the credibility of the WGCNA, each group should have more than 10 samples; 3) the tissues gathered for sequencing should be peripheral blood mononuclear cells; and 4) the raw or processed data of these datasets are available to access [10]. Finally, the training dataset of normal controls and breast cancer patients in GSE27562 and GSE81622 were selected. The expression data were downloaded from the GEO via the "GEOquery" package, and then the probes were annotated with their gene symbols [11].

## WGCNA and identification of shared genes

We performed WGCNA on the Bioinfo Intelligent Cloud website to identify the coexpressed gene modules of SLE and BRCA and chose the modules that were significantly positively related to SLE and negatively related to BRCA [12]. The shared genes in the selected modules were obtained with a Venn diagram on the Bioinformatics & Evolutionary Genomics website.

## Expression and clinical correlation analysis

The RNA-seq data obtained from the TCGA were analyzed to compare the expression level of MME, the shared gene signature, in normal and cancer tissues, and then the results were visualized with "ggplot2" (version 3.3.3) in R language. Additionally, the UALCAN website was used to explore the protein expression level of MME in the CPTAC dataset [13]. The Kruskal–Wallis test was used for the correlation analysis between clinical features and MME expression levels [14].

## Construction of ROC curves

The "pROC" package (version 1.17.0.1) was employed to construct receiver operating characteristic (ROC) curves to test the sensitivity and specificity of diagnosis via the MME expression level. Ranging from 0.5 to 1, an area under the ROC curve (AUC) close to 1 represents flawless predictive ability.

## DNA methylation and transcription factor analysis

DNA methylation can regulate gene expression levels without altering the sequence. Thus, the MEXPRESS web server was chosen to explore the MME DNA methylation level of multiple probes in BRCA samples from TCGA datasets [15]. To explore the upstream regulator, we also used the TRRUST website to identify the transcription factors regulating the expression of MME [16].

## Functional enrichment analysis

With the STRING web server, we acquired the top 20 MME-related proteins [17]. Gene Ontology (GO) and Kyoto Encyclopedia of Genes and Genomes (KEGG) enrichment analyses were used to enrich potential pathways with the KOBAS web server [18], and then the results were visualized using the "ggplot2" package (version 3.3.3) in the R language.

## Identification of the target ncRNA and enrichment analysis

Using the miRNet website, we identified the potential upstream miRNAs of MME and the miRNA-related circRNAs. Then, from the webpage, we obtained the miRNA-circRNA networks and circRNAs with the top degrees that may be involved in the molecular pathways of

MME [19]. KEGG pathway analysis was performed with DIANA-miRPath v3.0 to explore their biological functions [20].

## Analysis of MME and survival, immunotherapy response and drug sensitivity

With the Kaplan–Meier plotter web server, survival plots were created to explore the associations between MME expression and relapse-free survival (RFS), postprogression survival (PPS), overall survival (OS), and distant metastasis-free survival (DMFS) in breast cancer, and the corresponding log rank P value was calculated to compare the difference in survival curves [21]. Then, the ROC plotter website was used to investigate the relationship between MME expression and the response to therapy of breast cancer patients [22]. The GSCA web server was employed to analyze the correlation between MME expression and the sensitivity (IC50) to 265 small molecules from the Genomics of Drug Sensitivity in Cancer (GDSC) database by Pearson correlation analysis [23].

## Tumor-infiltrating immune cell evaluation

The "GSVA" (gene set variation analysis) package in R language was employed to explore the associations between MME and 24 types of immune cells [24]. The following immune cell types were analyzed: macrophages, pDC (plasmacytoid dendritic cells), aDC (activated dendritic cells), iDC (immature dendritic cells), DC (dendritic cells), mast cells, eosinophils, NK CD56bright cells, NK cells, NK CD56dim cells, neutrophils, cytotoxic cells, B cells, T cells, CD8 + T cells, Treg cells, Tcm (T central memory) cells, follicular Tfh (T follicular helper cells), T helper (Th) cells, T gamma delta (Tgd) cells, T effector memory cells, Th1 cells, Th2 cells, and Th17 cells.

# Results

## WCGNA: The coexpression gene module

With the Spearman correlation coefficients, heatmaps were plotted to assess the correlation between each gene module and SLE or BRCA, and each color signifies a different gene module (Fig 1A and 1B). Three modules, "tan", "midnight blue", and "green", that were positively correlated with SLE were selected (tan module: r = 0.539, p = 0.00034; midnight blue module: r = 0.602, p = 4e−05; and green module: r = 0.623, p = 1.7e−05), and they included 280 genes overall. For BRCA, "red", "purple", and "green yellow" were negatively correlated with BRCA (red module: r = -0.854, p = 1.6e-06, purple module: r = -0.681, p = 0.00096, and green yellow module: r = -0.499, p = 0.025), and they included 233 genes overall (Fig 1C and 1D). The only overlapping gene between the selected modules positively correlated with SLE and negatively correlated with BRCA was *MME* (Fig 1E), which is expected to be related to the pathogenesis of both diseases.

## Association between MME expression and clinical outcomes

To analyze the potential role of MME in BRCA, we compared MME expression between tumor and matched normal tissues. The MME expression level was significantly decreased in tumor tissues at the mRNA and protein levels compared with the corresponding noncancerous tissues (S1A and S1B Fig), suggesting a potential role in the occurrence and development of tumors. ROC analysis was performed to test the diagnostic value of MME in BRCA cases. The AUC value of MME was 0.984 with a 95% confidence interval equaling 0.976–0.992, indicating

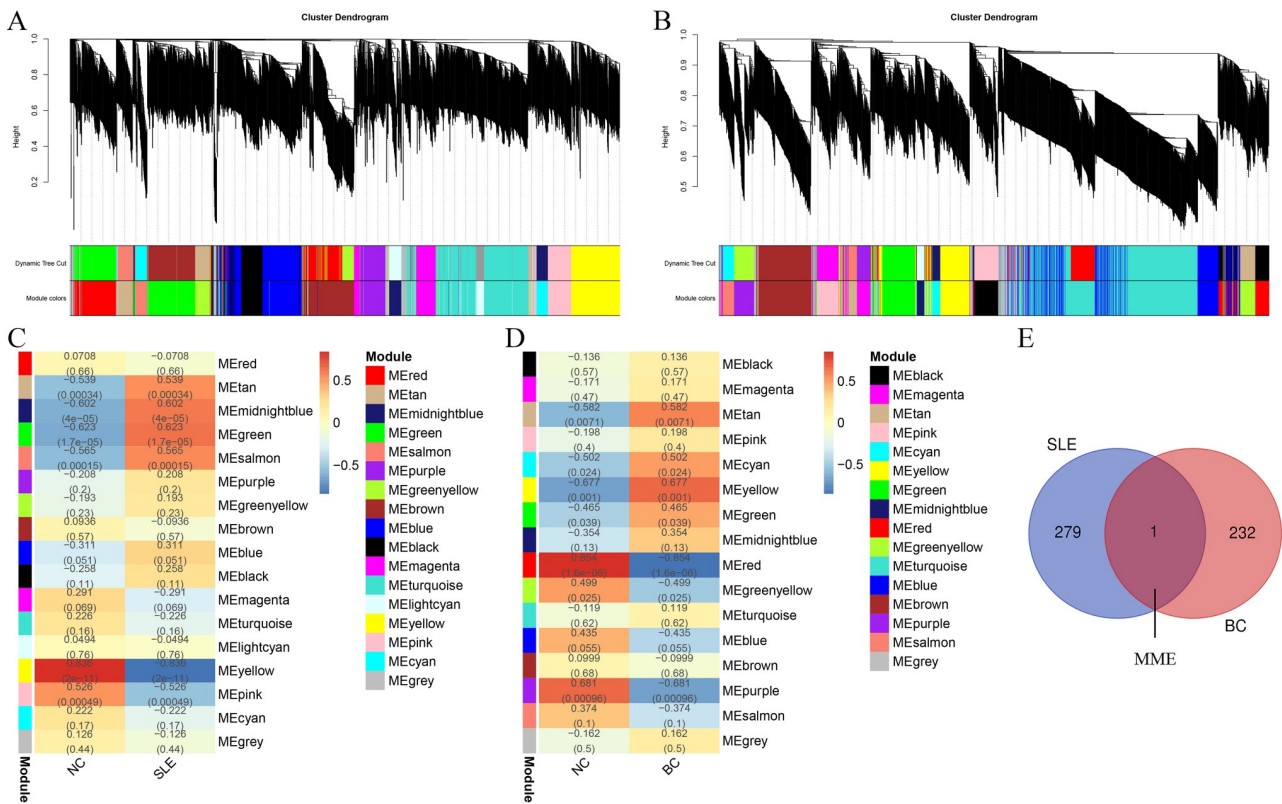

**Fig 1. WGCNA analysis of systemic lupus erythematosus (SLE) and breast cancer (BC).** (A) The cluster dendrogram of co-expression genes in SLE. (B) The cluster dendrogram of co-expression genes in BC. (C) Module–trait relationships in SLE. (D) Module–trait relationships in BC. (E) The common gene shared in SLE and BC.

the excellent diagnostic value of MME in BRCA (S1C Fig). In total, 1222 samples with expression profile and clinical feature data obtained from TCGA were included in the analysis.

Then, we downloaded data for BRCA cases from the TCGA and divided these cases into low and high expression groups according to the median MME expression level. Statistical analysis identified a significant association between high MME expression and normal tissue status (Fig 2A–2D and 2F) and age under 60 (Fig 2E), but there was no significant correlation between MME expression and different TNM or pathologic stages (Table 1). Interestingly, low MME expression was significantly correlated with the subtypes luminal B (LumB) (Fig 2G) and infiltrating ductal carcinoma (Fig 2H); thus, MME may serve as a specific biomarker for BRCA types. To further examine the role of MME in BRCA progression, we compared the MME expression level between primary breast tumor samples and bone metastasis samples (GSE146661) and lung or brain metastasis samples (GSE191230) with the Wilcoxon rank sum test. The results showed that MME expression was significantly decreased in the metastasis samples (Fig 2I and 2J), suggesting that MME may inhibit the proliferation and metastasis of BRCA.

## DNA methylation analysis

To investigate the potential regulatory mechanism of MME, we visualized the DNA methylation levels of MME with the MEXPRESS tool. Most probes showed a significantly negative association with MME mRNA levels (S2 Fig), suggesting that MME expression was

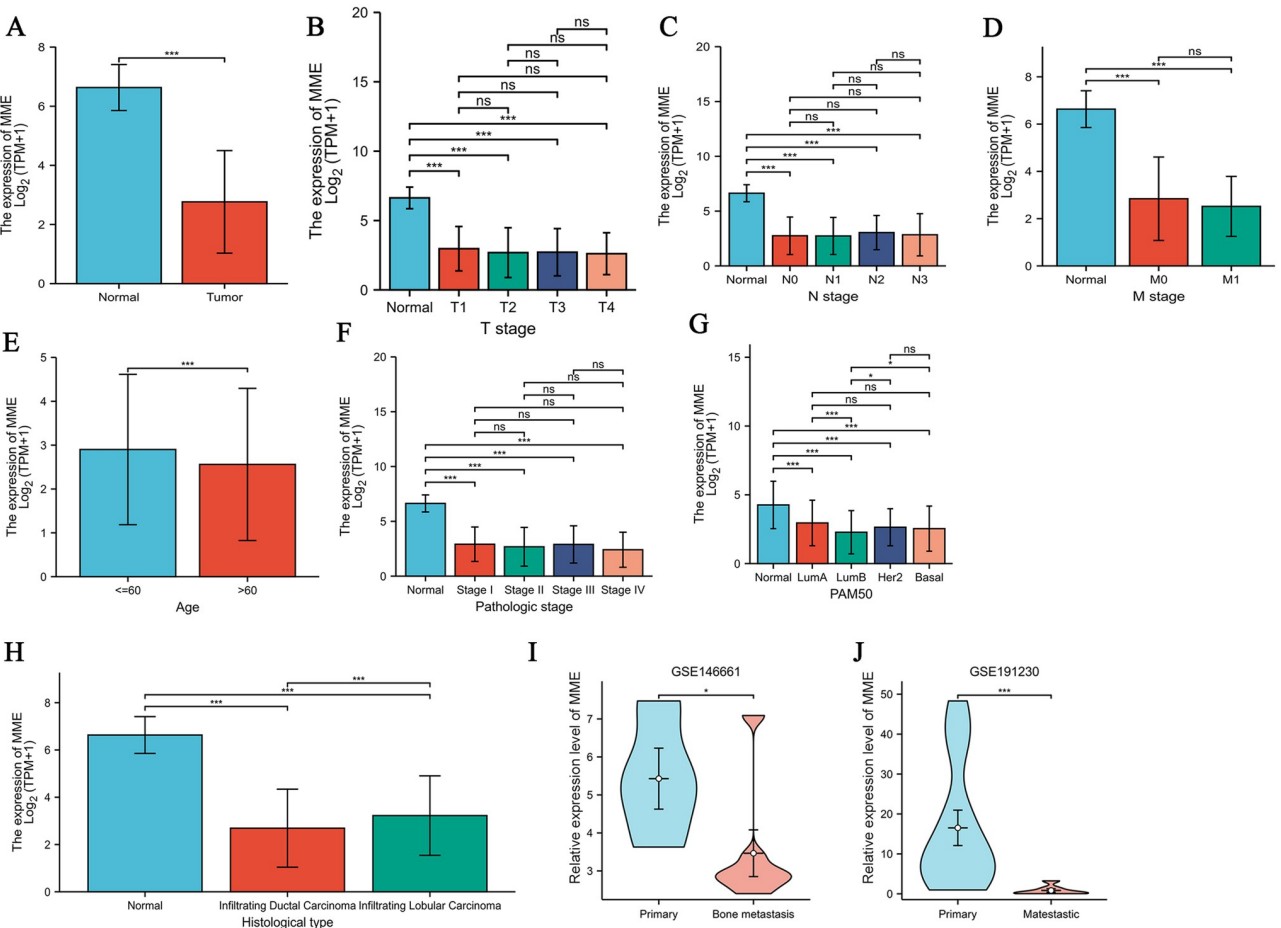

**Fig 2. MME expression in BRCA were compared in different clinicopathological parameters.** (A) MME expression between normal tissues and cancer tissues. (B) MME expression among different T stages. (C) MME expression among different N stages. (D) MME expression among different M stage. (E) MME expression between patients over or under age 60. (F) MME expression among different pathologic stages. (G) MME expression among patients with different PAM50 (prediction analysis of microarray50). (H) MME expression among different histological types. (I) MME expression between primary breast cancer and lung or brain metastasis samples. (J) MME expression between primary breast cancer and bone metastasis samples. $^{*}$p < .05, $^{**}$p < .01, and $^{***}$p < .001.

significantly decreased in tumor tissues and may be negatively regulated by DNA methylation. Moreover, MME expression was significantly related to histological type (p = 3.632e-13), HER2 receptor status (p = 4.560e-4), sample type (p = 6.152e-103), and subtype of BRCA (p = 4.432e-4).

## Enrichment analysis of MME-related proteins

The top 20 MME-binding proteins were obtained with the STRING tool (Fig 3A). Then, the protein–protein interaction (PPI) network of these selected proteins was drawn with the Cytoscape (Fig 3B) [25]. Next, GO and KEGG enrichment analyses for MME-related proteins were executed to obtain a better understanding of their biological functions and examine the potential relationship of MME with SLE and BRCA (Fig 3C and 3D). Enrichment analysis identified several pathways associated with tumorigenesis, such as the "FoxO (forkhead Box O) signaling pathway", "PI3k-Akt (phosphatidylinositol 3-kinase/Protein Kinase-B) signaling pathway", and "B-cell receptor signaling pathway". More detailed GO and KEGG

**Table 1. Clinicopathologic features of BRCA from the TCGA.**

| Characteristics | Low expression of MME | High expression of MME | p |
|---|---|---|---|
| **Total** | 543 | 544 | |
| **Age, n (%)** | | | **0.001** |
| < = 60 | 275 (25.3%) | 328 (30.2%) | |
| > 60 | 268 (24.7%) | 216 (19.9%) | |
| **PR status, n (%)** | | | **0.040** |
| Negative | 188 (18.1%) | 154 (14.8%) | |
| Indeterminate | 1 (0.1%) | 3 (0.3%) | |
| Positive | 327 (31.5%) | 365 (35.2%) | |
| **ER status, n (%)** | | | 0.778 |
| Negative | 124 (11.9%) | 116 (11.2%) | |
| Indeterminate | 1 (0.1%) | 1 (0.1%) | |
| Positive | 391 (37.6%) | 406 (39.1%) | |
| **HER2 status, n (%)** | | | 0.946 |
| Negative | 256 (35.1%) | 304 (41.7%) | |
| Indeterminate | 6 (0.8%) | 6 (0.8%) | |
| Positive | 73 (10%) | 84 (11.5%) | |
| **Pathologic T stage, n (%)** | | | 0.070 |
| T1 | 122 (11.3%) | 156 (14.4%) | |
| T2 | 326 (30.1%) | 305 (28.1%) | |
| T3&T4 | 92 (8.5%) | 83 (7.7%) | |
| **Pathologic N stage, n (%)** | | | 0.432 |
| N0 | 259 (24.3%) | 257 (24.1%) | |
| N1 | 182 (17%) | 177 (16.6%) | |
| N2 | 49 (4.6%) | 67 (6.3%) | |
| N3 | 38 (3.6%) | 39 (3.7%) | |
| **Pathologic M stage, n (%)** | | | 0.520 |
| M0 | 432 (46.7%) | 473 (51.1%) | |
| M1 | 11 (1.2%) | 9 (1%) | |

**Abbreviation**: MME: Membrane metalloendopeptidase; TCGA: The cancer genome atlas; T stage: Tumor stage; M stage: Metastasis stage; N stage: Node stage; PR: progesterone receptor; ER: estrogen receptor; HER2: human epidermal growth factor receptor 2.

enrichment analysis data are shown in S1 Table. With the KEGG pathway database, we further identified that the MME-binding proteins PTEN and BCL6 were involved in the FoxO signaling pathway, and PTEN and CD19 were involved in the PI3k-Akt signaling pathway.

## Prediction and enrichment analysis of target miRNAs, circRNAs and transcription factors

Considering the role of miRNAs and circRNAs in gene expression regulation and cancer progression [26], we predicted the potential MME-related miRNAs and miRNA–target circRNAs in BRCA tissues to construct a miRNA-circRNA network. Finally, we obtained 10 potential upstream miRNAs of MME, of which the interaction networks are shown in S3A Fig. Then, we screened 14 potential miRNA–target circRNAs with the "Minimum Network" tool to reduce the number of circRNAs to obtain the most relevant circRNAs, and the miRNA-circRNA network is displayed in S3B Fig. The degree and betweenness of the selected miRNAs and circRNAs in the miRNA-circRNA network are shown in S2 Table. Then, the identified

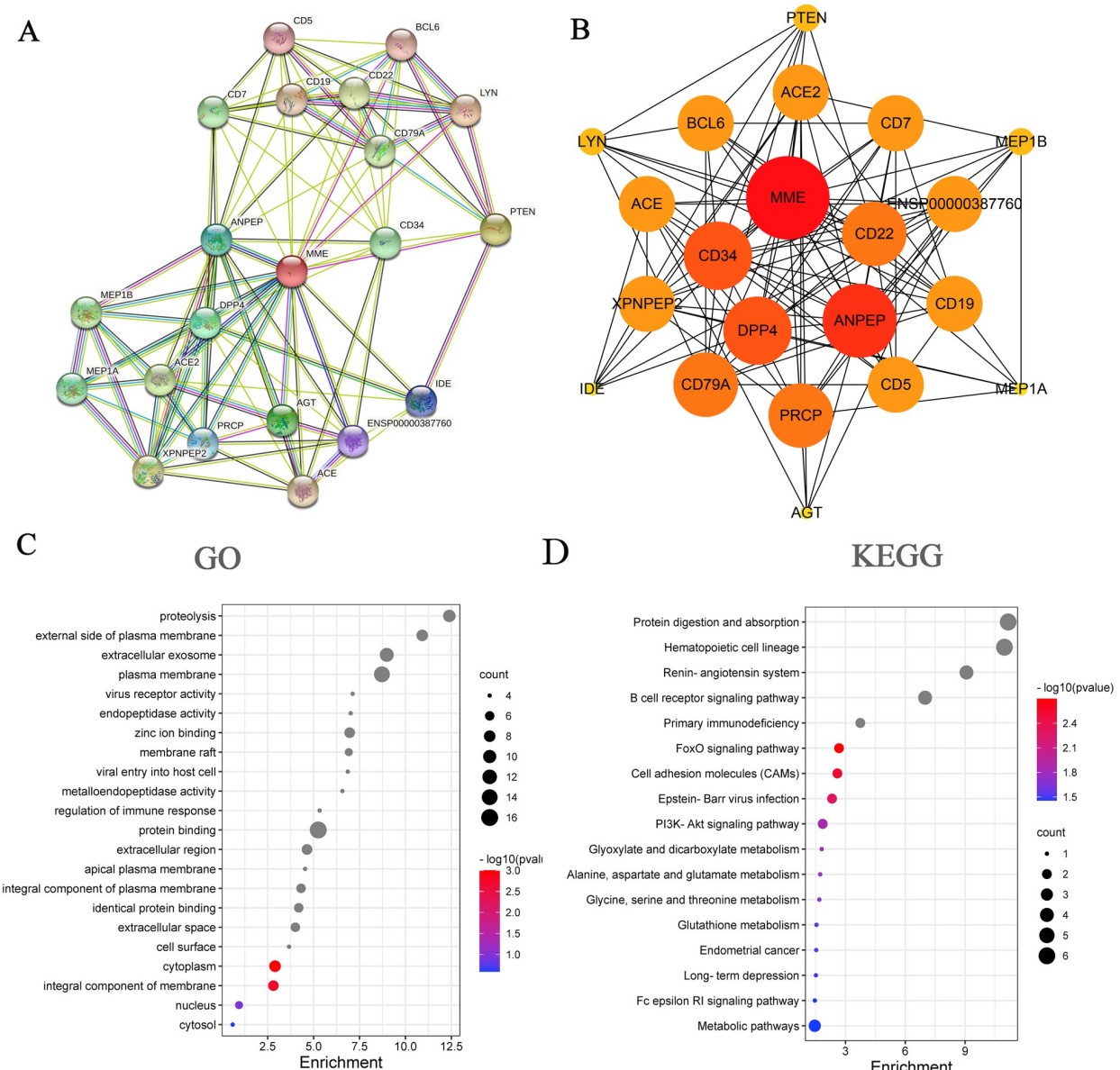

**Fig 3. MME-related gene enrichment analysis.** (A) The top 20 available experimentally determined MME-binding proteins obtained from STRING website. (B) The protein-protein interaction (PPI) network of the MME-binding proteins. (C) GO analysis of the MME-binding and interacted proteins. (D) KEGG pathway analysis of the MME-binding and interacted proteins. The color from blue to red was correlated with -log(p value).

miRNAs were further explored by KEGG enrichment analysis. The results revealed that these miRNAs were involved in multiple biological processes, such as ECM-receptor interaction, amoebiasis, and fatty acid biosynthesis. Interestingly, the KEGG results for miRNAs and MME-related proteins showed enrichment of the "FoxO signaling pathway" and "PI3k-Akt signaling pathway" in both analyses, indicating the important role of the PI3K/AKT/FOXO signaling pathway in the pathogenesis of SLE and BRCA (S3C Fig). More detailed results of the enrichment analysis are shown in S3 Table. To explore the upstream regulator of MME, we employed the TRRUST website, which revealed HOXC6, MYC, SP1, and SPI1 as the transcription factors that regulate the expression of MME (S4 Table).

### Survival, immunotherapy response and drug sensitivity analysis

The survival analysis revealed that low MME expression was related to favorable RFS (P<0.05) in BRCA but not other clinical outcomes (Fig 4A–4D). A low MME expression level was related to resistance to chemotherapy in BRCA. Moreover, low MME expression was associated with resistance to chemotherapy for several GDSC small molecules across cancers. The association between MME expression and sensitivity to the top 30 GDSC drugs across cancers is shown in Fig 4C, and more data are shown in S5 Table.

### Analysis of immune infiltration

According to the median MME expression level, the selected samples were divided into high and low expression groups. The infiltration levels of a total of 19 types of immune cells were affected by the expression level of MME, including T cells, Th1 cells, Th2 cells, TFH cells, Tgd cells, Tcm cells, Tem cells, T helper cells, neutrophils, mast cells, NK cells, NK CD56dim cells, macrophages, iDCs, eosinophils, DCs, cytotoxic cells, CD8 T cells, and B cells (S4A Fig). Among the 24 types of immune cells, the levels of 20 types were significantly positively correlated with MME expression levels, while Th2 cells were the only type that was significantly negatively correlated with MME (S4B Fig). Macrophages and neutrophils had the most significant positive correlation with MME expression (S4C and S4D Fig).

### Discussion

Despite decades of research, the BRCA incidence continues to rise and remains the leading cancer type in women, affecting one in 20 women worldwide and up to one in eight women in

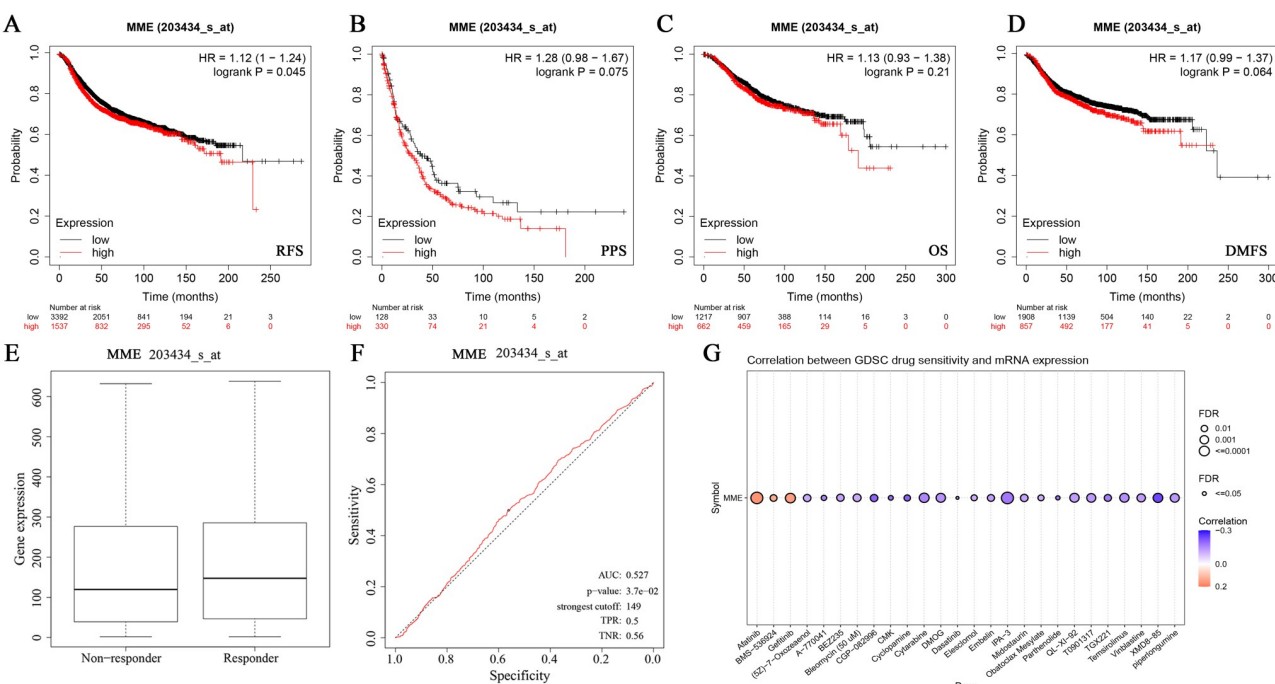

**Fig 4. Analysis of MME expression of survival, immunotherapy response and drug sensitivity.** (A) The correlation of MME expression and RFS. (B) The correlation of MME expression and PPS. (C) The correlation of MME expression and OS. (D) The correlation of MME expression and DMFS. (E-F) The ROC plot of the association between MME expression and the response to therapy of BRCA. (G) Bubble plot of the correlation between MME mRNA expression and GDSC drug sensitivity for the top 30 ranked drugs. The color from blue to red represents the correlation between mRNA expression level and drug sensitivity (IC50, 50%inhibiting concentration).

high-income countries [27, 28]. Surprisingly, several cohort studies and meta-analyses revealed that BRCA incidence is decreased in SLE patients vs. the general population, which inspired our interest in exploring the potential pathways involved in the reduced incidence of BRCA in SLE patients [2]. Therefore, we used WGCNA to identify the crosstalk of SLE- and BRCA-related molecular mechanisms and revealed that MME was positively correlated with SLE but negatively correlated with BRCA.

The *MME* gene, located on human chromosome 3q21-27, encodes a 100-kD transmembrane glycoprotein, and its active site is located in the extracellular environment [29]. MME is usually downregulated in cancers and serves as a tumor suppressor. For example, downregulated MME was correlated with advanced tumor stage, and overexpression of MME interrupted the FAK-RhoA axis to inhibit tumor cell metastasis in esophageal squamous cell carcinoma [8]. In prostate carcinogenesis, MME is downregulated in nearly 50% of cancers and inhibits carcinogenesis by regulating the activity of prostate stem/progenitor cells in cooperation with PTEN [9]. Similar to the abovementioned tumors, MME expression levels were significantly decreased in BRCA tumor tissues compared with corresponding noncancerous tissues. Moreover, downregulation of MME markedly correlated with the BRCA subtypes LumB and infiltrating ductal carcinoma. ROC curve analysis indicated the potential diagnostic value of MME in BRCA, with an AUC value equal to 0.984 (95% CI = 0.976–0.992). Low MME expression was associated with favorable RFS (P<0.05) but was also associated with resistance to chemotherapy (P<0.05) in BRCA.

To explore the molecular mechanisms responsible for the reduced morbidity of BRCA in SLE patients, we obtained MME-related proteins and miRNAs and then performed GO and KEGG enrichment analyses. Finally, the PI3K/AKT/FOXO signaling pathway was identified as a potential target mechanism that reduced the incidence of BRCA in SLE patients with both MME-related proteins and miRNAs. The PI3K-AKT pathway is one of the most commonly activated cancer drivers induced by activated membranous receptor tyrosine kinases to promote tumor cell survival, proliferation, growth, and angiogenesis in human cancers [30]. Activated PI3K induces the transformation of PIP2 (phosphatidylinositol 4,5-bisphosphate) to PIP3 (phosphatidylinositol 3,4,5-trisphosphate), and then PIP3 activates downstream protein kinases. Activated by PIP3, AKT regulates cellular metabolism partly via downstream mTOR (mechanistic target of rapamycin) and FoxO transcription factors [31]. The PI3K-AKT-mTOR signal transduction pathway is a major molecular mechanism involved in cellular resistance to extracellular stimulation, including various growth factors and inflammatory cytokines, and it has an important influence on tumor cell survival, proliferation, and growth [32]. Recognized as the dominant oncogenic mechanism, PI3K/Akt/mTOR is activated in most breast cancers, and inhibitors of this pathway exhibit therapeutic potential in clinical practice [33]. Commonly considered tumor suppressors, FoxO family members have a conserved DNA-binding domain and play a highly cell-type-specific role in oxidative stress resistance, cell cycle progression, apoptosis, and differentiation by entering the nucleus and triggering the transcription of various genes [34]. However, activated AKT inhibits the translocation of FoxO1 from the cytosol to the nucleus by stimulating the phosphorylation of FoxO1, which is then ubiquitinated and degraded by proteasomes [35]. As MME was significantly downregulated in tumor tissues and is involved in the PI3K/Akt pathway, we suggest that MME may act as a tumor suppressor in breast cancer by regulating the PI3K/Akt signaling pathway, especially its downstream FoxO signaling pathway. Moreover, the mechanism responsible for metastasis inhibition by MME seems to be associated with decreased activation of FAK, RhoA, and downstream MEK/ERK, which are involved in tumor migration and metastasis [8]. However, due to the limited current research, the specific mechanism of MME in BRCA still needs further study.

Due to the moderating role of the mRNA–miRNA-circRNA network in tumorigenesis and tumor growth, we considered that noncoding RNAs (ncRNAs) might also participate in MME-mediated regulatory mechanisms in BRCA. In our study, we identified hsa-miR-1-3p and NBPF9 as the most important miRNA and circRNA, respectively. miRNAs alter the expression of oncogenic or tumor-suppressive target genes to participate in the pathogenesis of cancers [36]. By targeting certain genes, hsa-miR-1-3p represses the invasion and growth of tumor cells [37]. CircRNAs, namely, miRNA sponges, can bind miRNAs and then inhibit their expression to competitively suppress their interaction with target mRNAs, indirectly regulating the expression of certain genes and finally modulating tumor progression [38]. Above all, we considered that miRNAs and circRNAs regulate the tumorigenesis of BRCA in part by controlling MME expression.

Immune infiltration may also be involved in the occurrence and growth of BRCA, and MME most significantly correlates with macrophages and neutrophils. Macrophages are the most common tumor-associated stromal cells in the tumor microenvironment, and their phagocytosis leads to tumor elimination, inflammatory activation, and antigen presentation, thereby inducing adaptive immunity to tumors [39, 40]. Neutrophils, recognized as inflammatory immune cells, can function as tumor suppressive regulators via neutrophil extracellular trap formation. This extracellular fiber network negatively impacts surrounding cells due to the high local concentrations of a toxic mixture of nuclear DNA and granule proteins induced by neutrophil DNA [41].

We identified the commonly dysregulated molecular mechanisms in both SLE and BRCA, which may partly explain the low risk of BRCA in SLE patients and offer novel methods for the prophylaxis and treatment of BRCA. The limitations of this study should be addressed. We assessed expression levels in PMBCs, which may not fully reflect the expression profile of BRCA. In addition, these findings should be validated and explored in in vitro or in vivo experiments to draw a more reliable conclusion.

## Conclusion

In summary, we identified that MME was positively related to SLE but negatively related to BRCA. MME expression is decreased in breast tumor tissues and was identified as a diagnostic biomarker for BRCA with high accuracy. Low MME expression was related to better RFS and resistance to chemotherapy. The PI3K/AKT/FOXO signaling pathway could be dysregulated and reduce BRCA risk in SLE patients. The crosstalk between MME and the PI3K/AKT/FOXO signaling pathway in BRCA needs further investigation.

## Supporting information

**S1 Fig. Expression analysis of MME in tumor tissues and normal tissues.** (A) The expression level of MME mRNA between tumor tissue and normal tissue of BRCA. *** P < .001. (B) The expression level of MME total protein between tumor tissue and normal tissue of BRCA. **** P < .0001. (C) ROC analysis was performed to examine the diagnostic value of MME. (TIF)

**S2 Fig. Association between MME and DNA methylation.** The beta value of methylation, the Benjamini-Hochberg-adjusted P-value and the Pearson correlation coefficients (R) are displayed. (TIF)

**S3 Fig. MME-related miRNA and circRNA prediction and KEGG enrichment analysis.** (A) The potential upstream miRNA of MME. (B) The network of MME-related miRNA-circRNA.

(C) KEGG enrichment analysis of the potential upstream miRNAs. PI3K/AKT and FOXO signaling pathways were highlighted as the common pathways consistent with the KEGG analysis of MME-related proteins.
(TIF)

**S4 Fig. Analysis of the immune infiltration of 24 types of immune cells.** (A) Enrichment scores of immune cells in low and high MME expression group. (B) Lollipop plot illustrated the Spearman's correlation and p-value of all immune cells. (C) Scatter plot exhibited the Spearman's correlation and p-value of macrophages. (D) Scatter plot exhibited the Spearman's correlation and p-value of neutrophils.
(TIF)

**S1 Table. KEGG and GO analysis of MME-related proteins.**
(DOCX)

**S2 Table. The degree and betweenness of the selected miRNA and circRNA in the miRNA-circRNA network.**
(DOCX)

**S3 Table. KEGG and GO analysis of MME-related miRNAs.**
(DOCX)

**S4 Table. The transcription factors regulating MME.**
(DOCX)

**S5 Table. The correlation between GDSC drug sensitivity and MME mRNA expression in pan-cancer.** IC50 of 265 small molecules in 860 cell lines and its corresponding MME mRNA gene expression was obtained from GDSC.
(DOCX)

## Author Contributions

**Conceptualization:** Jiatong Ding, Wenxiong Zhang.

**Data curation:** Jiatong Ding, Chenxi Li, Kexin Shu, Wanying Chen, Chenxi Cai, Xin Zhang, Wenxiong Zhang.

**Formal analysis:** Jiatong Ding, Wenxiong Zhang.

**Funding acquisition:** Wenxiong Zhang.

**Investigation:** Jiatong Ding, Xin Zhang, Wenxiong Zhang.

**Methodology:** Jiatong Ding, Chenxi Li, Wanying Chen, Chenxi Cai, Wenxiong Zhang.

**Project administration:** Wenxiong Zhang.

**Resources:** Jiatong Ding, Wenxiong Zhang.

**Software:** Jiatong Ding, Chenxi Li, Kexin Shu, Wanying Chen, Chenxi Cai, Xin Zhang, Wenxiong Zhang.

**Supervision:** Wenxiong Zhang.

**Validation:** Wenxiong Zhang.

**Visualization:** Wenxiong Zhang.

**Writing – original draft:** Jiatong Ding, Xin Zhang, Wenxiong Zhang.

**Writing – review & editing:** Jiatong Ding, Wenxiong Zhang.

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
