## [Decision Letter · Decision Letter 0]

20 Mar 2023

PONE-D-22-34693

Membrane metalloendopeptidase (MME) is positively correlated with systemic lupus erythematosus and may inhibit the occurrence of breast cancer

PLOS ONE

Dear Dr. Zhang,

Thank you for submitting your manuscript to PLOS ONE. After careful consideration, we feel that it has merit but does not fully meet PLOS ONE’s publication criteria as it currently stands. Therefore, we invite you to submit a revised version of the manuscript that addresses the points raised during the review process.

We look forward to receiving your revised manuscript.

Kind regards,

Samikshan Dutta, Ph.D.

Academic Editor

PLOS ONE

Journal Requirements:

Reviewers' comments:

Reviewer's Responses to Questions

**Comments to the Author**

1. Is the manuscript technically sound, and do the data support the conclusions?

Reviewer #1: Partly

Reviewer #2: Partly

2. Has the statistical analysis been performed appropriately and rigorously? 

Reviewer #1: Yes

Reviewer #2: I Don't Know

3. Have the authors made all data underlying the findings in their manuscript fully available?

Reviewer #1: Yes

Reviewer #2: Yes

4. Is the manuscript presented in an intelligible fashion and written in standard English?

Reviewer #1: Yes

Reviewer #2: Yes

5. Review Comments to the Author

Reviewer #1: The manuscript by Ding et al. reports that Membrane metalloendopeptidase (MME) is positively correlated with systemic lupus erythematosus and may inhibit the occurrence of breast cancer. Authors also identified that MME is a highly accurate diagnostic biomarker for breast cancer. This is an excellent insight that will pique the readers' interest. However, few suggestions will increase the quality of manuscript.

1. The author should demonstrate the interaction between MME and the PI3K/AKT/FOXO signaling pathway in BRCA.

2. Authors should clarify the relationship between MME and all the different types of breast cancer.

3. According to this study, in Conclusion section author need to re-write this sentence “In summary, we identified that MME was negatively related to SLE but positively related to BRCA.”

4. Figure quality is very bad. Authors should improve.

5. There are couple of grammatical and spelling mistakes. Minor changes to the manuscript's writing will also help to reinforce the paper.

Reviewer #2: The manuscript by Ding et al revealed an interesting rationale and finding to define aggressive BC pathology. Based on Weighted gene coexpression network analysis between BC dataset and systemic lupus erythematosus dataset, the auhtrs group identified membrane metalloendopeptidase hold the key in BC pathology. They have extensively performed bioinformatics analysis to give a strong rationale and reason t believe MME plays a major role in BC.

As such the manuscript is interesting; however, the authors need to extensively address the manuscript.

Some of the major comments are

Though, we understood, majority of the data were analyzed using primary BC gene sequences and other parameters. It is important to confirm the hypothesis using any dataset from advanced BC or metastatic tissue derived gene sequences.

The second major concern is the manuscript did not address to show the relation between the original function of MME upon the adverse pathology conditions. It Is important that being peptidases, there might be numerous substrates for MME, the manuscript should address the relevance in this manuscript.

The methodology should be clear and brief. For example ¬ “Additionally, we visited the UALCAN website (http://ualcan.path.uab.edu/analysis-prot.html) to explore the protein expression level of MME in the CPTAC dataset [13]”. The website url details may not needed as it is abbreviated, and the detail should be give which dataset and cancer is analyzed

Then, we visited the ROC plotter website (http://www.rocplot.org/) to investigate the relationship between MME expression and the response to therapy of breast cancer patients [22].

The rationale for this analysis DNA methylation, methyltransferases and mismatch repair correlation. further the correlation though significant, not very strong, it is better be removed from this manuscript unless there is strong relevance.

In addition to the miRNA, TF analysis should give better evidence on the upstream regulator of MME. The manuscript should include the TF analysis.

6. PLOS authors have the option to publish the peer review history of their article (what does this mean?). If published, this will include your full peer review and any attached files.

Reviewer #1: No

Reviewer #2: No

---

## [Author Response · Author response to Decision Letter 0]

26 Apr 2023

Dear Editors and Reviewers:

Thank you for the letter and for the reviewers’ comments concerning our manuscript, entitled “Membrane metalloendopeptidase (MME) is positively correlated with systemic lupus erythematosus and may inhibit the occurrence of breast cancer” (ID: PONE-D-22-34693). All of the authors read the peer reviewers’ comments carefully and provided point-by-point response. All of the comments were valuable and very helpful for revising and improving our paper and provided important guiding significance of our research. We have substantially revised the manuscript according to the reviewers' comments. Revised portions are marked in red in the paper. All things considered, the main corrections in the paper and the responses to the reviewers’ comments follow.

Responses to the reviewer’s comments:

Reviewer #1:

1. Comment: The author should demonstrate the interaction between MME and the PI3K/AKT/FOXO signaling pathway in BRCA.

Response and changes: Thank you for your advice. We explored the roles of PTEN, BCL6, and CD19, the MME-binding proteins, in the FOXO and PI3K-AKT signaling pathways with the KEGG database in “Results” section. Although there is no evidence that MME is directly involved in these two signaling pathways, we believe that MME may regulate them indirectly through the interaction of the abovementioned proteins.

2. Comment: Authors should clarify the relationship between MME and all the different types of breast cancer.

Response and changes: Thank you for your comments. In Figure 2G-J, we added the expression of MME in different histological types of breast cancer and metastases.

3. Comment: According to this study, in Conclusion section author need to re-write this sentence “In summary, we identified that MME was negatively related to SLE but positively related to BRCA.”

Response and changes: Thank you for pointing out our clerical error; we have now corrected it in “Conclusion” section.

4. Comment: Figure quality is very bad. Authors should improve.

Response and changes: Thank you for pointing this out. We have remade the figures.

5. Comment: There are couple of grammatical and spelling mistakes. Minor changes to the manuscript's writing will also help to reinforce the paper.

Response: Thank you for pointing out our shortcomings, and we have carefully revised the text.

Thank you for the comments and suggestions.

Reviewer #2:

1. Comment: Though, we understood, majority of the data were analyzed using primary BC gene sequences and other parameters. It is important to confirm the hypothesis using any dataset from advanced BC or metastatic tissue derived gene sequences.

Response and changes: Thank you for your comments. In Figure 2I and J, we compare the expression of MME between primary breast cancer and brain, lung, and bone metastases from breast cancer.

2. Comment: The second major concern is the manuscript did not address to show the relation between the original function of MME upon the adverse pathology conditions. It Is important that being peptidases, there might be numerous substrates for MME, the manuscript should address the relevance in this manuscript.

Response: Thank you for pointing out our shortcomings. MME may act as a tumor suppressor in breast cancer by regulating the PI3K/Akt signaling pathway, especially its downstream FoxO signaling pathway. Moreover, the mechanism responsible for metastasis inhibition by MME seems to be associated with decreased activation of FAK, RhoA, and downstream MEK/ERK, which are involved in tumor migration and metastasis1. However, due to the limited current research, the specific mechanism of MME in BRCA still needs further study.

Changes: We have further discussed the mechanism of MME in the “Discussion” section. However, due to the limited number of current studies, it is difficult to clearly explain the detailed mechanism of MME. It is hoped that our research will arouse the interest of the academic community in MME, which will help us better answer this question.

3. Comment: The methodology should be clear and brief. For example ¬ “Additionally, we visited the UALCAN website (http://ualcan.path.uab.edu/analysis-prot.html) to explore the protein expression level of MME in the CPTAC dataset [13]”. The website url details may not needed as it is abbreviated, and the detail should be give which dataset and cancer is analyzed

Then, we visited the ROC plotter website (http://www.rocplot.org/) to investigate the relationship between MME expression and the response to therapy of breast cancer patients [22].

Response and changes: Thank you for your suggestions. We have simplified the “Method” section to avoid unnecessary descriptions.

4. Comment: The rationale for this analysis DNA methylation, methyltransferases and mismatch repair correlation. further the correlation though significant, not very strong, it is better be removed from this manuscript unless there is strong relevance.

Response and changes: Thank you for your suggestion; we have deleted these unimportant results.

5. Comment: In addition to the miRNA, TF analysis should give better evidence on the upstream regulator of MME. The manuscript should include the TF analysis.

Response and changes: Thank you for your advice. We supplemented this part of the results in the "Results" section. With the TRRUST website2, we identified HOXC6, MYC, SP1, and SPI1 as transcription factors that regulate the expression of MME (Table S4).

Thank you for the comments and suggestions.

We tried our best to improve the manuscript. These changes did not influence the framework of the paper. We did not list all changes here but marked them in red in the revised manuscript.

We earnestly appreciate the editor’s/reviewers’ hard work and hope that the corrections meet with approval.

Once again, thank you very much for the comments and suggestions.

Reference

1. Li M, Wang L, Zhan Y, Zeng T, Zhang X, Guan XY, et al. Membrane Metalloendopeptidase (MME) Suppresses Metastasis of Esophageal Squamous Cell Carcinoma (ESCC) by Inhibiting FAK-RhoA Signaling Axis. Am J Pathol. 2019; 189(7):1462-1472.

2. Han H, Cho JW, Lee S, Yun A, Kim H, Bae D, et al. TRRUST v2: an expanded reference database of human and mouse transcriptional regulatory interactions. Nucleic Acids Res. 2018 Jan 4;46(D1):D380-D386.

---

## [Decision Letter · Decision Letter 1]

31 Jul 2023

Membrane metalloendopeptidase (MME) is positively correlated with systemic lupus erythematosus and may inhibit the occurrence of breast cancer

PONE-D-22-34693R1

Dear Dr. Zhang,

We’re pleased to inform you that your manuscript has been judged scientifically suitable for publication and will be formally accepted for publication once it meets all outstanding technical requirements.

Kind regards,

Samikshan Dutta, Ph.D.

Academic Editor

PLOS ONE

Additional Editor Comments (optional):

Reviewers' comments:

Reviewer's Responses to Questions

**Comments to the Author**

1. If the authors have adequately addressed your comments raised in a previous round of review and you feel that this manuscript is now acceptable for publication, you may indicate that here to bypass the “Comments to the Author” section, enter your conflict of interest statement in the “Confidential to Editor” section, and submit your "Accept" recommendation.

Reviewer #1: All comments have been addressed

Reviewer #3: All comments have been addressed

2. Is the manuscript technically sound, and do the data support the conclusions?

Reviewer #1: Yes

Reviewer #3: (No Response)

3. Has the statistical analysis been performed appropriately and rigorously? 

Reviewer #1: Yes

Reviewer #3: (No Response)

4. Have the authors made all data underlying the findings in their manuscript fully available?

Reviewer #1: Yes

Reviewer #3: (No Response)

5. Is the manuscript presented in an intelligible fashion and written in standard English?

Reviewer #1: Yes

Reviewer #3: Yes

6. Review Comments to the Author

Reviewer #1: The authors have addressed all my comments for this paper. The paper has been significantly improved after revising. I will recommend the editor accept this paper and publish it.

Reviewer #3: This manuscript describes the in-silico validation of association of membrane metalloendopeptidase with SLE mediated inhibition of breast cancer. Since this is revision, I have focussed mainly on whether the authors have addressed the concerns raised by the primary reviewers. All the comments have been addressed and I belive manuscript is ready for publication.

7. PLOS authors have the option to publish the peer review history of their article (what does this mean?). If published, this will include your full peer review and any attached files.

Reviewer #1: No

Reviewer #3: No

---

## [Editor Report · Acceptance letter]

3 Aug 2023

PONE-D-22-34693R1 

Membrane metalloendopeptidase (MME) is positively correlated with systemic lupus erythematosus and may inhibit the occurrence of breast cancer 

Dear Dr. Zhang:

I'm pleased to inform you that your manuscript has been deemed suitable for publication in PLOS ONE. Congratulations! Your manuscript is now with our production department. 

Kind regards, 

on behalf of

Dr. Samikshan Dutta 

Academic Editor

PLOS ONE